# Evaluating the Efficacy of Common Treatments Used for *Vairimorpha (Nosema)* spp. Control

**Cody Prouty [1],\*, Cameron Jack [1] , Ramesh Sagili [2] and James D. Ellis [1]**

1   Entomology and Nematology Department, University of Florida, Gainesville, FL 32611, USA
2   Department of Horticulture, Oregon State University, Corvallis, OR 97331, USA
\*   Correspondence: cprouty@ufl.edu

**Abstract:** *Vairimorpha* (formerly *Nosema*) *apis* and *V. ceranae* are microsporidian pathogens that are of concern for managed honey bee colonies. Multiple treatments have been proposed to be effective in reducing the prevalence and intensity of *Vairimorpha* spp. infections. Here, we test the efficacy of these products in one lab-based experiment and three field experiments. In the lab experiment, we found no reductions in *Vairimorpha* spp. prevalence (proportion of individuals infected with *Vairimorpha* spp.) or intensity (number of *Vairimorpha* spp. spores per individual), but we did find a decrease in honey bee survival after treatment with Fumagilin-B, Honey-B-Healthy®, and Nozevit Plus. The first field experiment showed increased *Vairimorpha* spp. intensity in colonies treated with Fumagilin-B and HiveAlive® compared to a negative control (sucrose syrup alone). The second field experiment showed a weak reduction in *Vairimorpha* spp. intensity after 3 weeks post treatment with Fumagilin-B compared to Nozevit. However, *Vairimorpha* spp. intensity returned to levels comparable to those of other treatment groups after 5 weeks post treatment and remained similar to those of other groups for the duration of the experiment. The final field trial showed no positive or negative effects of treatment with Fumagilin-B or Nosevit on *Vairimorpha* spp. prevalence or intensity. These findings raise questions regarding the efficacy of the products currently being used by beekeepers to control *Vairimorpha* spp. We argue that the observed reduction of *Vairimorpha* spp. is more likely relevant to the phenology of spore prevalence and intensity in honey bee colonies than to chemical treatment.

**Keywords:** honey bee; *Vairimorpha* spp.; Fumagilin-B; HiveAlive®; Nozevit; treatment; prevalence; intensity



## 1. Introduction

The western honey bee, *Apis mellifera*, is an economically important pollinator in agricultural systems and faces stressors such as pathogens, parasites, and pests [1]. *Vairimorpha* (formerly *Nosema*, [2]) spp. are microsporidian parasites that reproduce in the midgut of honey bees and have been shown to be detrimental to honey bee health [3,4]. Infections with *Vairimorpha* spp. alter behavior, reduce immunity to other pathogens, increase energetic demand, and lead to increased mortality of individuals bees [5,6]. To date, three species of *Vairimorpha* have been found to infect honey bees; *V. apis*, *V. ceranae*, and *V. neumanni* [7–9]. These species differ in virulence, geographic range, and clinical signs associated with infection in honey bees [10].

In North America, fumagillin is the only approved antibiotic treatment for *Vairimorpha* spp. infections [11]. Fumagillin is a non-specific antibiotic derived from *Aspergillus fumigatus* [12]. Historically, a product called Fumagilin-B (Medivet Pharmaceuticals Ltd., High River, AD, Canada) was the only fumagillin product used in North America. In 2018, Medivet ceased production of this treatment, and Fumadil-B (KBNP, Inc., Anyang, Republic of Korea) is now the commonly used fumagillin-based treatment. While fumagillin products have been shown by some to be effective at reducing *Vairimorpha* spp. prevalence and

intensity [13,14], others have shown that treatment with fumagillin does not always control *Vairimorpha* spp. infection [15–18].

Some researchers have proposed that proper honey bee nutrition can prevent negative impacts of *Vairimorpha* spp. in honey bee colonies. Colonies with a high pollen diet have been shown to cope with *Vairimorpha ceranae* infection better than colonies that consume less pollen [19,20]. Furthermore, nutritional stress can change honey bee gut microbiota and suppress immune function, thus favoring *V. ceranae* infection [21]. Manufacturers of certain natural plant products, such as Nozevit (Apivita, Varaždin, Croatia) and Honey-B-Healthy® (Honey-B-Healthy, Inc, Cumberland, MD, USA), claim that treatments improve honey bee nutrition and ultimately colony survival following *Vairimorpha* spp. infection [17,22]. Similarly, another feeding supplement called HiveAlive® (Advance Science Ltd., Galway, Ireland) is purported to support good nutrition and intestinal well-being of honey bees, thereby reducing *V. ceranae* infection over time [23]. Many beekeepers throughout the world feed Nozevit, Honey-B-Healthy®, and HiveAlive® to their honey bee colonies several times each year with the intention of fortifying their immune response and protecting them from mortality caused by *Vairimorpha* spp.

In the present study, we tested *Vairimorpha* spp. infection control in honey bees by treatments (both registered and unregistered for *Vairimorpha* spp. control) commonly used by beekeepers. We conducted four separate experiments to test the efficacy of these treatments. In the first experiment, we compared the prevalence and intensity of *V. ceranae* and the subsequent survival of caged honey bees exposed to *V. ceranae* and treated with Fumagilin-B, Nozevit Plus, and Honey-B-Healthy®. In the second experiment, honey bee colonies were treated during the winter with Fumagilin-B and HiveAlive® and sampled monthly until spring. In the third and fourth experiments, colonies were treated during the winter with Nozevit Plus and Fumagilin-B following label recommendations for either "fall" or "spring" treatments. The objective of these experiments was to evaluate the efficacy of different treatments to reduce the prevalence (proportion of individuals infected with *Vairimorpha* spp.) and intensity (number of *Vairimorpha* spp. spores per individual) of *Vairimorpha* spp. infection, and to improve the health of honey bee colonies.

## 2. Materials and Methods

### 2.1. Experiment 1: Laboratory Cage Study for Evalting Efficacy of Fumagilin-B, Nozevit Plus, and Honey-B-Healthy® against Vairimorpha ceranae

In July 2014, capped brood combs were obtained from honey bee colonies at Oregon State University apiaries (Corvallis, OR, USA). We placed the combs in an incubator under simulated hive conditions (33 °C, 55% RH) to facilitate adult worker bee emergence. Twenty-four hours later, we gently brushed newly emerged bees into a large container and mixed them gently by hand. After the bees were mixed, we placed 250 individual bees inside cylindrical wire cages (63.06 cm$^3$) and returned them to the incubator, as per [19]. The caged bees were immediately provided with ad libitum access to a glass vial containing 25 mL of a *V. ceranae* spore/50% sucrose solution dosed at 40,000 spores/bee. The vials were covered with two layers of cheesecloth and then secured, inverted, and placed upon the top of the cage. Each cage also contained 25 g of finely ground wildflower pollen mixed with a 33% sucrose solution in a 2:1 (weight/volume) ratio. Prior to the experiment, the wildflower pollen was sent to the USDA National Science Laboratory (Gastonia, NC, USA) for pesticide analysis to assess the possible presence of pesticide residues. We found 34 ppb of fluvalinate, 3.7 ppb of chlorfenopyr, 2.6 ppb of trifluralin, and trace concentrations of all other pesticides that were tested for. The results of this pesticide residue analysis are presented as Supplementary Table S1 in the Supplemental Materials.

The spore concentration of the *V. ceranae* solution was formulated and purified by centrifugation following the methods of [24]. Briefly, the contents of infected gut samples were collected and centrifuged at 5000 rpm for five minutes at room temperature to produce a pellet of spores. After discarding the supernatant, the pellet was resuspended in distilled water by vortexing. This process was repeated 2–3 times. DNA analysis was performed



using the methods of [25] to confirm that only *V. ceranae* spores were present in the inoculum. Once the 25 mL solution containing *V. ceranae* was completely consumed, the caged bees were provided ad libitum access to water and 50% sucrose syrup (weight/volume). Three days after inoculation of bees with *V. ceranae* spores, we provided 25 mL of sucrose syrup containing one of four treatments to the cages. There were five cage replicates per treatment for the following four treatments: Fumagilin-B, Nozevit Plus, Honey-B-Healthy®, and a negative control (provided only 50% sucrose syrup). We prepared all treatments in 50% sucrose syrup according to the product label. Once the bees completely consumed the 25 mL of treated sucrose syrup, the caged bees were again given ad libitum access to water and sucrose syrup. In all, there were four treatments and five replicates (cages) of each treatment, for a total of 20 cages.

Once a week, we measured consumption of the pollen by bees and replaced unconsumed pollen with fresh pollen mixed with sucrose solution as described previously. Bee mortality was recorded every other day and dead bees were removed at the time of diet replacement for convenience. We measured consumption of both water and sucrose solution and replaced them on alternate days. At 16 and 23 days after spore inoculation, we removed 25 bees at random from each experimental cage for infection analysis. The abdomens of the bees were used to estimate *V. ceranae* prevalence (presence or absence of spores) and intensity (# of spores/bee) by light microscopy techniques as described by [26]. Each bee abdomen was checked individually for *V. ceranae* infection.

### 2.2. Experiment 2: Field Treatment Using HiveAlive® and Fumagilin-B

In December 2017, 90 colonies located in Waldo, Florida were identified and grouped into three apiaries of 30 colonies per apiary. As honey bee colony dynamics can influence *Vairimorpha* spp. prevalence and intensity [27], we ensured that all 90 colonies were of similar size and strength through equalization of the number of brood and adult bees prior to study initiation. These colonies were maintained by a local commercial beekeeper following best management practices common in the region (swarms controlled, pests managed, fed when necessary, etc.). Thirty colonies received HiveAlive® according to the product label. Another thirty colonies were treated with Fumagilin-B according to the product label. The remaining 30 colonies belonged to the negative control group and received untreated sugar syrup. All colonies were fed at the same time and the same amount of syrup, with at least 4 L of syrup per seasonal feeding.

On the day of first treatment, baseline data from all colonies were collected by sampling for *Vairimorpha* spp. and *Varroa destructor* (a parasitic mite of honey bees). Samples of adult bees were collected by shaking brood frames with nurse-aged bees from the combs onto a pan. About 300 bees were poured from the pan into sampling jars containing 70% ethanol. Later, *Vairimorpha* spp. intensity was estimated using the methods described by [27] from 100 bee pooled sub-samples. The infestation of *V. destructor* was determined as described by [28]. *Vairimorpha* spp. and *V. destructor* levels were monitored for every colony every 4–6 weeks from December 2017–May 2018.

### 2.3. Experiments 3 and 4: Winter Field Treatment Using Fumagilin-B and Nozevit at Fall and Spring Recommended Treatments

In January 2009 and December 2009 for Experiment 3 and 4 respectively, honey bee colonies in Umatilla, Florida and Windsor, Florida were assessed for the presence of *Vairimorpha* spp. infection.

### 2.4. Colony Selection

In Experiment 3, colonies were first randomly assigned to treatment groups, and then sampled for *Vairimorpha* spp. infection. The intensity of *Vairimorpha* spp. was not different among colonies or between treatments at the beginning of the experiment. In Experiment 4, we chose colonies that had a mild to moderate *Vairimorpha* spp. infection (~50,000 to 500,000 spores per bee) as candidates for the study. Of these positively infected

candidates, 50 queen right colonies of similar colony strength were selected and divided into the five treatment groups. All hives were equalized for honey stores within the supers. Each plot of ten hives was located in an open sunny field and placed on wooden pallets, approximately 15 m apart from other plots of colonies.

### 2.5. Vairimorpha spp. Treatments and Experimental Design

Negative control treatment groups received only sucrose syrup at each feeding application. We created two treatments for Nozevit and Fumagilin-B (Medivet Pharmaceuticals Ltd., Alberta Canada). We mixed dosages according to product labels from both products and made applications two (equivalent to a labeled "spring" treatment) and four (equivalent to a labeled "fall" treatment) times, seven days apart from one another. The two applications ("spring" treatment) were followed by two feedings of only sugar syrup when the "fall" treatment group received its third and fourth treatment. In total, we used five treatments in both experiments: a negative control, "spring" treatments (two applications) of Nozevit and Fumagilin-B, and "fall" treatments (four applications) of Nozevit and Fumagilin-B.

### 2.6. Sampling for Vairimorpha spp. Infection

For each sample of bees for *Vairimorpha* spp. assessment, we collected 100 bees per hive and placed them in 100 mL containers with 70% ethanol and returned them to the laboratory. There, we removed the abdomens of 100 bees per sample and combined them with 100 mL distilled water. We placed the solution into a sterilized Cooks Power Blender (JC Penny, Manchester, CT, USA) and blended for 30 s until an even suspension was formed. *Vairimorpha* spp. intensity was estimated using the methods described [26] from 100 bee pooled sub-samples. Using the 5-square method reported by [26], we calculated the number of spores per bee for each sample. In Experiment 3, we sampled baseline *Vairimorpha* spp. levels, again one week after the second treatment of all colonies, then one week after the fourth treatment of applicable colonies, and finally, three weeks after the fourth treatment. In Experiment 4, we again sampled *Vairimorpha* spp. at a baseline, followed by one week post treatment, two weeks post treatment, and a final sampling four weeks after the initial treatment.

### 2.7. Colony Assessment

The colony assessors were blind to the treatment group assignments and had not visited the site during treatment application. In Experiment 3, two observers estimated the area covered by bees and brood of all combs in the hives at the end of the experiment, and the average estimation of the observers was calculated [29]. In Experiment 4, a baseline and final assessment were made by one assessor.

### 2.8. Statistical Analyses

Statistical analyses for all experiments were performed using R version 4.1.1. The effect of time was analyzed as a factor in all analyses where applicable. In Experiment 1, we used linear models (normal error structures) to test for relationships between the *Vairimorpha* spp. treatment and average bee consumption of pollen, water, and sucrose using the following model structure in the lme4 package [30]: [response variable = treatment * time]. To compare bee survival across treatments, we used a Cox proportional hazards model in the survival package [31]. Bees that survived until the end of the experiment (Day 28) and those that were removed for *V. ceranae* intensity and prevalence were treated as censored cases.

To test the effects of Fumagilin-B and HiveAlive® on *Vairimorpha* spp. intensity, we used a linear mixed model in the lme4 package [30] using the log spore load + 1 as the response variable. Prevalence of *Vairimorpha* spp. and *V. destructor* in Experiment 2 were analyzed using general linear mixed models with binomial error distributions. For all response variables in Experiment 2, the error structures were: [response variable = treatment * time + colony (random effect)]. In Experiments 3 and 4, we analyzed the effect of the treatments (Fumagilin-B at fall and spring treatments, Nozevit at fall and spring treatments, and

the negative control group) on estimations of bees and brood using linear mixed models (normal error structures). The prevalence of *Vairimorpha* spp. was analyzed using general linear mixed models with the error structures: [response variable = treatment * time + colony (random effect)]. *Vairimorpha* spp. intensity (log + 1 transformed) in Experiments 3 and 4 was analyzed using linear mixed models with the same error structures as prevalence. Tukey's honest significant difference (HSD) was used on the models where multiple comparisons were made.

## 3. Results

### 3.1. Experiment 1: Laboratory Cage Study Evaluating Available Treatments against Vairimorpha ceranae

#### 3.1.1. Consumption of Pollen, Sucrose Solution, and Water

On average, bees consumed 18.4 mL of sucrose syrup, 43.2 mL of water, and 3.9 g of pollen over the course of 28 days. There was no evidence that the consumption of pollen ($F_{3,233} = 0.09$; $p = 0.967$), sucrose syrup ($F_{3,233} = 0.35$; $p = 0.791$) or water ($F_{3,233} = 0.48$, $p = 0.699$) was significantly different between bees in the various treatment groups. Trace amounts of several pesticides were found in the wildflower pollen that was provided to bees in experimental cages (Supplementary Table S1).

#### 3.1.2. *V. ceranae* Prevalence and Intensity

There was no evidence for an effect of an interaction between treatment and time on *V. ceranae* prevalence ($\chi^2 = 3.33$; df = 3; $p = 0.344$; Table 1) or for differences between treatments ($\chi^2 = 5.2$; df = 3; $p = 0.158$) or time ($\chi^2 = 0.51$; df = 1; $p = 0.475$). There was no evidence of significant differences in intensity of *V. ceranae* infection between the treatments ($F_{3,34} = 0.66$; $p = 0.586$). The median *V. ceranae* infection intensity for Honey-B-Healthy® treatment was $2.8 \times 10^6$ spores/bee, while Nozevit Plus, Fumagilin-B, and negative control treatments had a median infection intensity of $2.22 \times 10^6$, $1.13 \times 10^6$, and $1.83 \times 10^6$ spores/bee, respectively. There was evidence that *V. ceranae* intensity increased over time ($F_{3,36} = 4.21$, $p = 0.049$), but there was no evidence of an interaction between treatment and time of sampling ($F_{3,36} = 1.06$, $p = 0.38$).

**Table 1.** Experiment 1: Prevalence (percent of infected individuals) of *Vairimorpha* spp. in honey bees provided various treatments. The sampling occurred on two days (16 and 23) post treatment. Prevalence means with different letters are significantly different at $\alpha \leq 0.05$.

| Treatment | Sampling Day | Prevalence (%) |
|---|---|---|
| Control | 16 | 2.4 a |
| Fumagilin-B | 16 | 2.4 a |
| Honey-B-Healthy® | 16 | 7.2 a |
| Nozevit Plus | 16 | 6.4 a |
| Control | 23 | 4.0 a |
| Fumagilin-B | 23 | 2.4 a |
| Honey-B-Healthy® | 23 | 2.4 a |
| Nozevit Plus | 23 | 3.2 a |

#### 3.1.3. Survival Analysis

Kaplan–Meier survival curves were used to plot the survival data (Figure 1) and a Cox proportional hazards model was used to compare the survival curves of the various treatments. The Cox proportional hazards model indicated that there was evidence for reduction in survival among bees that were fed with all treatments compared to the negative control group ($\chi^2 = 10.8$; df = 3; $p = 0.013$). Kaplan–Meier curves showed that bees in the negative control group had the greatest survival, followed by that of bees in the Nozevit Plus and Honey-B-Healthy® group, and the Fumagilin-B treatment group (Figure 1).

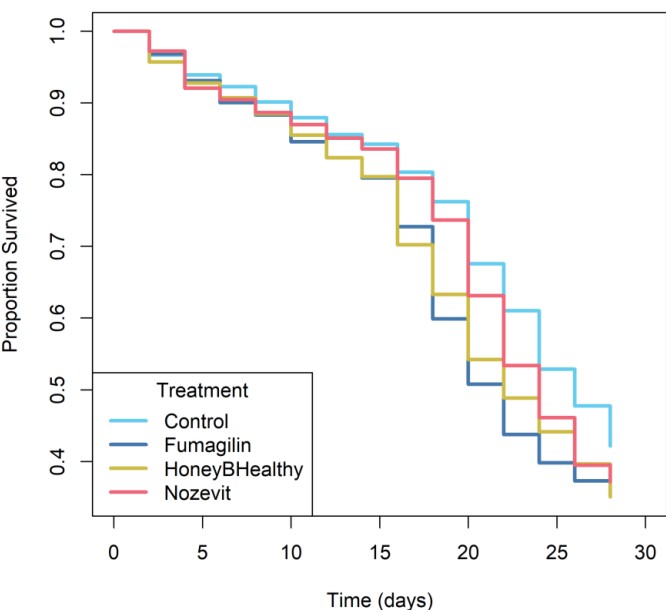

**Figure 1.** Proportion of honey bees that survived *Vairimorpha ceranae* infection and subsequent treatment in Experiment 1. Treatments are negative control (light blue), Fumagilin (dark blue), Honey-B-Healthy (yellow), and Nozevit Plus (red).

*3.2. Experiment 2: Field Experiment—HiveAlive® and Fumagilin-B*

Overall, there was strong evidence that the number of *V. destructor* per 100 bees across all treatments decreased over the course of the experiment ($\chi^2_{4,439} = 42.19$, $p < 0.001$), and there was an effect of treatment ($\chi^2_{2,439} = 7.87$, $p = 0.02$), but there was no evidence of an interaction between treatment and time ($\chi^2_{8,439} = 5.49$, $p = 0.705$) (Figure 2). Multiple comparisons showed that HiveAlive had a lower prevalence of *V. destructor* than controls, and Fumagilin-B was not different from either one.

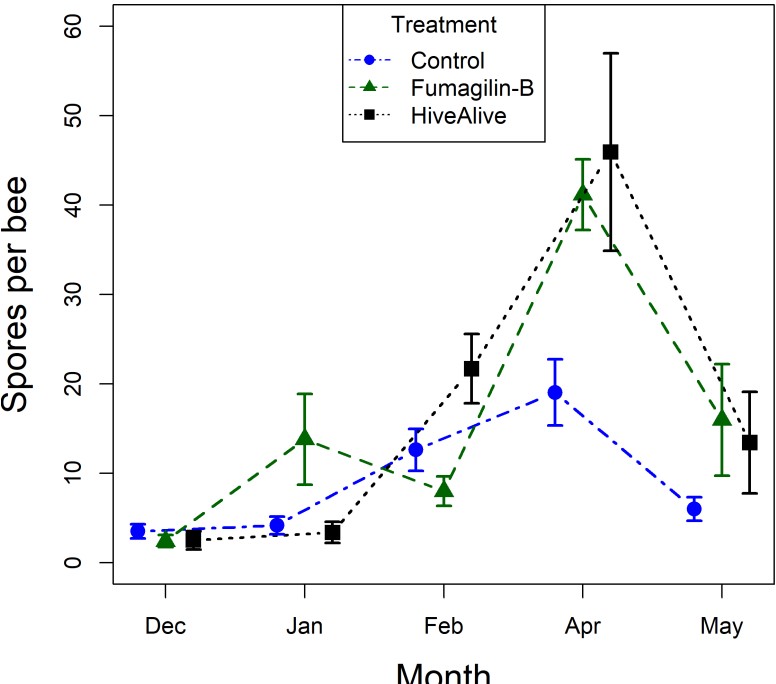

**Figure 2.** *Vairimorpha* spp. spores per bee (×50,000) in Experiment 2. Shapes and error bars represent mean and standard error. Treatments are: control (blue circles), Fumagilin-B® (green triangles), and Hive Alive (black squares).

Generally, *Vairimorpha* spp. intensity in bees increased over the course of the experiment, regardless of treatment, then decreased in the final month of the experiment. There was strong evidence of an interaction between treatment and time on *Vairimorpha* spp. intensity ($F_{8,438} = 2.88$, $p = 0.004$). In the multiple comparison breakdown, there was strong evidence of an increase in *Vairimorpha* spp. intensity in April for the bees fed the Fumagilin-B treatment ($z = 2.72$, $p = 0.006$) and the Hive Alive® treatment ($z = 3.27$, $p = 0.001$). In contrast, there was no evidence of a corresponding increase in intensity in bees in the negative control group (Figure 2).

### 3.3. Experiment 3: Winter Field Treatment using Fumagilin-B and Nosevit at Fall and Spring Recommended Treatments

At the end of the experiment, there was no evidence that colonies from the treatment groups differed in the observed number of bees ($F_{4,39} = 0.32$, $p = 0.866$) or brood ($F_{4,39} = 0.73$, $p = 0.58$) compared to those of the negative control.

At the end of the experiment, there was an average *Vairimorpha* spp. spore load of 516,666.67 spores per bee, with a prevalence of 0.84 across all treatments. There was no evidence that *Vairimorpha* spp. intensity was affected by treatment ($F_{4,45} = 0.69$, $p = 0.604$), but there was very strong evidence that *Vairimorpha* spp. intensity was affected by time ($F_{3,131} = 24.39$, $p < 0.001$). There was no evidence of an interaction between treatment and time on *Vairimorpha* spp. intensity ($F_{12,127} = 1.22$, $p = 0.279$). Similarly, there was very strong evidence that *Vairimorpha* spp. prevalence was only affected by the time variable ($\chi^2 = 24.27$, $p < 0.001$), with no differences between treatments ($\chi^2 = 4.17$, $p = 0.384$); Figure S1.

Reducing treatments to just the ingredient (negative control, Fumagilin-B, and Nozevit), regardless of solution preparation, showed weak evidence of an interaction between the treatment and time ($F_{6,136} = 2.07$, $p = 0.056$) on *Vairimorpha* spp. intensity. Again, there was strong evidence of an effect of time, where *Vairimorpha* spp. intensity dropped after the first week in all treatments ($F_{3,136} = 10.79$, $p < 0.001$). There was no effect of treatment ($F_{2,46} = 1.39$, $p = 0.259$).

### 3.4. Experiment 4: Repeat of Winter Field Treatment using Fumagilin-B and Nozevit at Fall and Spring Recommended Treatments

Overall, the number of observed bees and brood increased over time (bees: $F_{1,93} = 125.05$, $p < 0.001$; brood: $F_{1,93} = 181.67$, $p < 0.001$), but there was no evidence of an effect from treatments on these variables (bees: $p = 0.279$; brood: $p = 0.564$). For both bees and brood, there was no evidence of an interaction between treatment and time (bees: $p = 0.257$; brood: $p = 0.584$).

Across all treatments and observations, there was an average *Vairimorpha* spp. intensity of 673,500 spores per bee, with a prevalence of 0.8. There was no evidence of an effect of treatment ($F_{2,195} = 5.8$, $p = 0.214$) on *Vairimorpha* spp. intensity or the interaction between treatment and time ($F_{12,195} = 14.32$, $p = 0.281$), but very strong evidence that *Vairimorpha* spp. intensity was affected by time ($F_{3,195} = 40.39$, $p < 0.001$). Multiple comparisons showed an increase in *Varirmorpha* spp. intensity from week 1 to 2, then intensity decreased again after week 2. For *Vairimorpha* spp. prevalence, there was again no evidence for an effect of treatment $\chi^2 = 7.33$, $p = 0.119$) or the interaction between treatment and time ($\chi^2 = 10.43$, $p = 0.578$), but there was very strong evidence of an effect of time ($\chi^2 = 26.17$, $p < 0.001$).

## 4. Discussion

Many consider the control of pathogens such as *Vairimorpha* spp. essential for honey bee health [11] and important for reducing yearly honey bee colony losses. Our results provide weak or no evidence of any available registered or unregistered treatments to reduce *Vairimorpha* spp. prevalence and intensity. Laboratory exposure of bees to *Vairimorpha* spp. and subsequent treatment with various therapeutics showed no evidence of treatment effects on *Vairimorpha* spp. spore prevalence or intensity (Experiment 1). Subsequent field experiments showed increased *Vairimorpha* spp. intensity following exposure to HiveAlive® and Fumagilin-B (Experiment 2), weak short-term benefits of Fumagilin-B (Experiment 3),

or no effects of treatments at all (Experiment 4). In all experiments, time was the principal factor that predicted *Vairimorpha* spp. infection.

Fumagilin-B, Honey-B-Healthy®, and Nozevit treatments reduced the survival of bees when compared to that of bees fed only sugar water (the negative control group). We speculate that, in a colony setting, bees would likely interact with treatments mixed into supplemental sugar syrup differently than they would in the laboratory cage setting. In cages, for instance, honey bees are forced to feed on the provided treatments (no choice scenario), thus maximizing the treatment received per bee. In the field, however, honey bees would have access to ample pollen, honey, and nectar stores, in addition to supplemental sucrose. Therefore, caged bees (in a no choice/forced feeding scenario) might be exposed to higher amounts of treatment than bees in the field colonies, which could lead to reductions in honey bee survival. As for Fumagilin-B specifically, [12] reviewed negative effects of the compound on honey bee health and found fumagillin, along with its often overlooked counterion (dicyclohexylamine) in salt form, to be toxic to honey bees. More recently, researchers have found significant reductions in mortality following treatment with dicyclohexylamine in isolation; this counterion is present in fumagillin-DCH salt and in Fumagilin-B [32].

In the first field experiment (Experiment 2), bees treated with Fumagilin-B and HiveAlive® showed increased *Vairimorpha* spp. intensity in April compared to that of bees in the negative control group. Previous studies have found that the death of beneficial gut bacteria resulting from exposure to antibiotics leads to increased vulnerability to *Vairimorpha* spp. infection and antibiotic resistant pathogens in honey bees [33–35]. Given that the gut microbial communities of honey bees have high specificity and stability [34], it could take time for populations of beneficial microbes to rebuild post treatment, allowing opportunistic parasites to become established in the meantime. Thus, if antibiotic treatments become necessary, subsequent probiotic treatments and protein nutrition may also be essential to prevent establishment of parasites e.g., [36–38]. We hypothesize that this disruption of gut microbial communities led to higher *Vairimorpha* spp. intensity in the treatment groups seen in this study.

The *Vairimorpha* species dominant in each experiment could influence the efficacy of the treatments used in these experiments. *Vairimorpha ceranae* and *V. apis* have been shown to differ in their respective biology and their impacts on honey bee colonies [10,39,40]. In Experiment 1, we confirmed *V. ceranae* as the dominant species of *Vairimorpha* via molecular detection. However, the species of *Vairimorpha* was not confirmed in Experiment 2. Nevertheless, this experiment was performed between late 2017 and early 2018. At that time, *V. ceranae* was the dominant species in honey bees throughout most of the world [40]. Experiments 3 and 4 were performed in early 2009 and from late 2009–early 2010, respectively. During these years, *V. ceranae* was present but likely not the dominant species. There may have been a mix of *V. apis* and *V. ceranae* infection in bees in these studies [10].

*Vairimorpha apis* infections in honey bees have a predictable phenology [40–43], where infections generally decrease in the summer and increase throughout fall and winter, with a peak in early spring [44,45]. This seasonality was particularly evident in Experiment 2, where a large peak was seen in April, particularly in colonies treated with Fumagilin-B and HiveAlive® (Figure 2). In contrast, *V. ceranae* has been shown to be present year-round [46,47] and have higher thermal tolerance than *V. apis* [48–50]. While *V. ceranae* appears not to have an easily predictable phenology, it can have periods of high infection followed by periods of low infection prevalence and intensity [51]. Thus, optimal management of honey bee colonies with the intention to outlast periods of high *V. ceranae* infection may be more effective than relying on treatments to reduce infection.

Proper nutrition appears to be effective for mitigating the effects of *Vairimorpha* spp. infection. In general, increased pollen consumption leads to improved outcomes following *Vairimorpha* spp. infection [19,20,52–54]. Despite this, it is unclear if pollen substitutes are as effective as natural pollen consumption by honey bees for the reduction of spore loads [reviewed in 54]. In periods of pollen dearth, certain pollen substitutes would likely

improve outcomes for bees with *Vairimorpha* spp. infection [55,56], but might also lead to higher *Vairimorpha* spp. intensities. It is difficult to recommend specific pollen substitutes since the ingredients used and the formulation processes in the different products are highly variable.

Although fumagillin and other treatments designed to reduce *Vairimorpha* spp. infections have been shown to be effective in certain circumstances, there are many factors that can affect colonies' responsiveness to treatments. Based on the findings described here, it is not feasible to recommend one specific treatment over another for *Vairimorpha* spp. control; rather, we recommend improving colony health and reducing pest pressure to ensure colonies are not afflicted with multiple stressors. Future research should focus on examining timing of treatments with predicted *Vairimorpha* spp. peaks. For this research, sampling for *Vairimorpha* spp. could be conducted throughout the year to predict when peaks would occur, as geography and climate play a significant role. Then, treatments could be applied the following year after understanding the phenology of *Vairimorpha* spp. Additionally, future research might further explore how certain products work on a preventative basis rather than a treatment basis, such as the use of propolis as a preventative treatment for *Vairimorpha* spp. infections [57]. Other preventative treatments that take *Vairimorpha* spp. seasonality into account could also prove to be effective. Such information would be valuable for both researchers and beekeepers.

**Supplementary Materials:** The following supporting information can be downloaded at: https://www.mdpi.com/article/10.3390/app13031303/s1, Figure S1: Number of spores per bee in Experiment 3; Table S1: Pesticide results.

**Author Contributions:** Conceptualization, C.P., C.J., R.S. and J.D.E.; Methodology, C.J., R.S. and J.D.E.; Formal analysis, C.P.; Investigation, C.J.; Resources R.S. and J.D.E.; Data curation, C.P., C.J. and J.D.E.; Writing—original draft, C.P.; Writing—review & editing, C.P., C.J., R.S. and J.D.E.; Visualization, C.P.; Supervision, C.J., R.S. and J.D.E.; Project administration, J.D.E.; Funding acquisition, C.J., R.S. and J.D.E. All authors have read and agreed to the published version of the manuscript.

**Funding:** This research was supported by the USDA National Institute of Food and Agriculture Multistate Project (1005822), the Florida State Beekeepers Association and the Florida Department of Agriculture and Consumer Services. The funding sources had no involvement in any aspects of the study.

**Institutional Review Board Statement:** Not applicable.

**Informed Consent Statement:** Not applicable.

**Data Availability Statement:** The data in this manuscript are available in a public repository at: https://doi.org/10.5281/zenodo.7534789.

**Acknowledgments:** We gratefully thank Straughn Farms and Billy Rhodes for providing many experimental honey bee colonies during our field experiments. Additionally, we thank Walter Taylor, Jon Elmquist, Branden Stanford, and Michelle Weschler for their help sampling experimental honey bee colonies and counting *Vairimorpha* spp. spores.

**Conflicts of Interest:** The authors declare no conflict of interest.

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
