# Peer review of "Evaluating the Efficacy of Common Treatments Used for Vairimorpha (Nosema) spp. Control"

_applsci, doi:10.3390/app13031303_

Round 1

Reviewer 1 Report

This study presents an evaluation of the efficacy of three common treatments used for Nosema control:  Fumagilin-B, Honey-B-Healthy® , and Nozevit Plus. Laboratory and field experiments were performed to assess the effectiveness of treatments to reduce prevalence and intensity of Nosema infection, maintaining and improving the health of the honey bees. None of the treatments tested proved effective in controlling the infection. Any changes observed in the number of spores were more related to the natural phenology of Nosema than to the effect of the treatments. Rather, the treatments had likely negative effects on bee survival in the laboratory. 

The experiments were carried out with proper methodology and using a large number of bees and hives. The results are clearly and fully presented, and the discussion is exhaustive. I believe this work has relevance for beekeepers who can choose between various commercial products for treatments based on scientific data.

I make only minor editing comments:

- the L for litre is sometimes written in lower case and other times in upper case;

- check that scientific names are always written in italics;

- be consistent with the use of bold type in paragraph titles.

Only one thing is not clear to me. One of the products tested is sometimes called HiveAlive and other times Honey-B-Healthy. Is it the same product with two names or is there a difference? 

Reviewer 2 Report

This manuscript present data regarding the analysis of Vairimorpha spp. prevalence, a microsporidian pathogen that causes infections in bees, and treatment with 3 products to control this pathogen. The paper discusses several interesting aspects related to importance to control this pathogen and the existent methods. The study was well done, and the results were clearly explained. The figures and tables are very clear, the images provided are resolutive and the captions are extremely informative. The techniques used were sufficient for that proposed by the authors. The paper is well written for most part. However some aspects about manuscript need to be revised.

I recommend the publication in Applied Sciences after minor corrections.

Specific comments

Material and Methods

1) The authors need to better explain how the spores were diluted.

2)   It was not well explained how the products were diluted and offered to the bees.

3)  Why was experiment 2 conducted with HiveAlive® and Fumagilin-B?

4)  Page 4: 2009 or 2019? If it was done before, it must come before. Explain these dates better or remove them from the text.

5)  Page 4: “We created two treatments for Nozevit and Fumagilin-B…” Were these treatments repeats or were they different concentrations? Explain.

6) Page 7: “Overall, there was strong evidence that the number of Varroa per 100 bees…”. Varroa? I didn't understand this term.
